# The Quality of Life of Patients with Surgically Treated Mandibular Fractures and the Relationship of the Posttraumatic Pain and Trismus with the Postoperative Complications: A Prospective Study

**DOI:** 10.3390/medicina55040109

**Published:** 2019-04-17

**Authors:** Tanja Boljevic, Batric Vukcevic, Zoran Pesic, Aleksandar Boljevic

**Affiliations:** 1Clinic of Otorhinolaryngology and Maxillofacial Surgery, Clinical Center Montenegro, 20000 Podgorica, Montenegro; boljevictanjamini@gmail.com; 2Faculty of Medicine, University of Montenegro, 20000 Podgorica, Montenegro; 3Department of Maxillofacial Surgery, Clinic of Dentistry, 18101 Nis, Serbia; pesic.z@gmail.com; 4Primary Health Care Center, 20000 Podgorica, Montenegro; boaleksandar@t-com.me

**Keywords:** mandible, fracture, questionnaire, quality of life, surgery

## Abstract

*Background and objectives:* Due to the fact that the mandible is the only movable bone in the face, it is often exposed to the influence of external forces. The incidence of trismus and posttraumatic pain in unilateral mandibular corpus fractures may be related to the occurrence of complications. There is a decrease in the quality of life of these patients. The aim was to study the relationship of the preoperative pain and trismus with the incidence of complications, as well as to investigate the quality of life. *Materials and Methods:* A prospective study on 60 patients with isolated mandibular fractures was performed, with a follow-up period of six months. The level of preoperative pain was measured on a 0–10 scale, while the mouth opening was measured with a caliper. All patients were treated surgically on the third day after the fracture. The University of Washington Quality of Life (UW-QOL v4) questionnaire was used to analyze the quality of life. *Results:* The most common types of complications were the occlusal derangement and facial asymmetry. The majority of complications were treated with counseling and physical therapy. The degree of preoperative pain was significantly positively related to the onset of complications (r_s_ = 0.782, *p* = 0.004). The interincisal distance showed a significant inverse relation with the incidence of complications (r_s_ = −0.722, *p* < 0.001). The patients regarded the pain, appearance and mood issues as the most important issues during the first postoperative month. *Conclusions:* The degree of inflammatory symptoms may be positively related to the onset of complications occurring after the rigid fixation of mandibular fractures. The postoperative health-related and overall quality of life was unsatisfactory in nearly half of the patients.

## 1. Introduction

Mandibular fractures are a common issue in maxillofacial surgery, with an increasing incidence [1]. Due to the fact that the temporomandibular joint is the only mobile joint in the skull, the mandible is often exposed to the influence of external forces—often leading to structural damage, which reflects on the aesthetics and the functionality of the face. Mandibular fractures may be associated with severe morbidity and serious consequences [2]. The majority of research on mandibular trauma is usually directed towards the epidemiology of mandibular fractures—and, in a lesser amount, the factors which influence the treatment success. The complexity of these injuries is seen in the different aspects of surgical treatment, the existence of various complications, as well as the influence on the general health and the integrity of the patient.

The advantage of the intraoral surgical approach lies in the absence of the cutaneous scars, as well as the superior maintenance of oral hygiene, speech, and feeding [3,4]. Several authors state that the optimal time to repair the facial fractures surgically is during the first 72 h—while others suggest that the surgery should be performed after several days, in order for the soft tissue swelling to recede [5]. Corticosteroids are frequently used with the aim to reduce the postoperative inflammation through their pharmacological effects (inhibition of vascular permeability and vasodilation, as well as the reduction of chemotaxis). Different administration routes have been proposed (intramuscular, intravenous, submucosal, intra-alveolar, etc.). A systematic review by Troiano et al. showed no difference in the postoperative pain, swelling, and trismus in patients who received dexamethasone via the submucosal or intramuscular route [6].

The mandibular fractures are associated with different preoperative, intraoperative and postoperative complications [5,7]. Some of the factors influencing the occurrence of complications are the patient’s age, the type and the site of the fracture, the general health and dentition, inadequate stabilization, treatment cost, and low postoperative compliance to the surgical advice [8]. The surgical treatment of the mandibular fractures is associated with a specifically strong sense of fear [9,10]. Therefore, the psychological consequences of the injury and the treatment should be addressed during the healing process as well [11].

The aim of the study was to determine the potential relationship of the preoperative factors (the degree of oral hygiene, pain, and interincisal distance) with the complications of isolated mandibular fractures which were treated surgically through the intraoral approach. Additionally, the quality of life of operated patients was estimated. Finally, a comparison was made between the patients with or without complications regarding their quality of life. These findings should spark interest in the risk factors for postoperative complications, as well as the psychological issues related to the mandibular fracture surgery.

## 2. Materials and Methods

### 2.1. Study Setting

The study was performed at the Clinic for Otorhinolaryngology and Maxillofacial Surgery of the Clinical Center of Montenegro. For this purpose, an analysis was made on all the patients with isolated mandibular fractures treated in the aforementioned institution during 2017. The patients were subject to a six-month follow-up after surgery.

### 2.2. Inclusion and Exclusion Criteria

The study was performed on patients of both genders aged 20–70 years suffering from an isolated mandibular corpus fracture which was treated with the intraoral surgical approach (reposition and osteosynthesis with titanium plates). The exclusion criteria were: Pregnancy, the presence of mental illness or alcohol/substance abuse, congenital osteological deformities, and acquired bone disorders and diseases.

### 2.3. Study Sample

The sample consisted of 51 male and nine female patients, and the demographic data (age and profession), as well as the degree of oral hygiene, preoperative pain, and trismus was collected.

### 2.4. Analysis Methods (Diagnosis, Oral HYGIENE, Pain, Trismus)

The diagnosis of unilateral mandibular corpus fracture was made by the anamnesis, clinical (inspection and palpation) and radiologic examination (X-ray, orthopantomography, and computed tomography). In order to examine the outcome of the treatment, preoperative and postoperative photographs were used, as well as study models. The degree of oral hygiene was estimated as poor, medium or good.

The third day after the injury, the degree of preoperative pain and trismus was examined. The mouth opening was measured between the superior and inferior incisives by caliper, and the results were expressed in millimeters. The pain was estimated on a 0–10 scale, with 0 being the painless state and 10 representing severe, unbearable pain.

### 2.5. Treatment

The surgery was performed on the third day after the injury under total anesthesia, with an intraoral incision at the level of the mandibular corpus, the reposition of bone fragments, and fixation with two titanium plates and screws. Intravenous clindamycin was administered (600 mg every 8 h), as well as the intravenous ketorolac. On the second postoperative day, the patients were discharged, with a prescribed treatment consisting of clindamycin tablets and oral analgesics (acetaminophen or ibuprofen) up to the tenth postoperative day. 

### 2.6. Follow-Up

The follow-up consisted of weekly exams during the first month, followed by visits after two, three and six months. Some patients were additionally examined by orthopantomography. The patient was subject to soft foods and strict oral hygiene during the first two postoperative months. After the first postoperative month, the patients were advised to perform mouth opening exercises, with a referral to physical rehabilitation if the patients encountered difficulties with mouth opening after two weeks of exercise. 

### 2.7. Quality of Life Analysis

The quality of life of the treated patients was assessed with the standardized University of Washington Quality of Life (UW-QOL v4) questionnaire, consisting of questions related to pain, appearance, activity, recreation, chewing, swallowing, speech, taste and salivation, as well as a specific segment related to the psychological issues (anxiety and mood disorders). The questionnaire was filled one month after the surgery.

### 2.8. Ethical Approval

All subjects gave their informed consent for inclusion before they participated in the study. The study was conducted in accordance with the Declaration of Helsinki, and the protocol was approved by the Ethics Committee and the Institutional Review Board of Clinical Center of Montenegro (decision No. 03/01—22513/1, approved on 26.12.2016.).

### 2.9. Statistical Analysis

The statistical analysis was performed in SPSS 21. The descriptive statistics protocol was used, as well as the chi-square test for comparison and the Spearman’s coefficient of correlation of nonparametric data. The *p*-value below 0.05 was considered statistically significant.

## 3. Results

The average age of the patients was 38.8 years, with a predominance of male patients (Table 1). The mandibular fractures were most common in ages 21–40 (*n* = 38, 63.3%). The most common causes of mandibular fractures were traffic accidents (43.3%), violence (38.3%), falls (10%), sports activities (6.6%) and other various reasons (1.8%). The fractures affected the left side of the mandibular corpus more commonly (35 cases, 58.3%) in contrast with the right side (25 cases, 41.7%). All fracture lines showed a pattern which was adequate for surgical treatment, thus enabling surgery in all cases.

Of all the 60 patients included in the study, nine patients (15%) suffered complications.

The relationship between pain intensity and trismus with the onset of complications is shown in Table 2. The majority of patients showed a low degree of preoperative pain. However, the degree of preoperative pain was positively related to the onset of complications, and this association was statistically significant (r_s_ = 0.782, *p* = 0.004). The majority of patients had a small interincisal distance (less than 5 mm). The interincisal distance showed a significant inverse relation with the incidence of complications (r_s_ = −0.722, *p* < 0.001).

Table 3 shows the distribution of different types of postoperative complications and the frequency of different treatment types. The most common types of complications were the occlusal derangement and facial asymmetry (together representing more than half of all complications). Other complications, such as an impaired mouth opening, malunion and nonunion, were less frequent. The majority of complications were treated with counseling and physical therapy. Nonunion was treated with a debridement and intermaxillary fixation, while malunion was treated with a refracture and repeated surgery. During the six-month follow-up, all the complications were treated successfully.

The degree of oral hygiene was not significantly related to the onset of complications (*p* = 0.58). It was estimated as good in the majority of patients (32 patients, 53.3%), out of which two patients (3.3%) had complications. Medium oral hygiene was seen in 21 patients (35%), out of which four patients (6.7%) had complications; and poor in seven patients (11.7%), where three patients (5%) suffered complications.

Table 4 contains the UW-QOL v4 domain scores of the patients included in the study. It can be seen that the lowest mean score was in the domain of chewing (50), while the highest score was in the domains of speech and taste (72.27), which seem to be affected the least. A higher degree of anxiety was seen in female patients in comparison with male patients, as well as in patients younger than 50 years of age. The anxiety extent was somewhat similarly distributed across the score levels.

The patients with complications showed a significantly lower score regarding appearance (*p* = 0.023), swallowing (*p* = 0.011) and anxiety (*p* < 0.01) compared to the uncomplicated patients. The other domain scores were not significantly different between the two groups.

Table 5 contains the answers to the global questions regarding the health-related quality of life compared with one month before the injury, as well as the health-related and overall quality of life during the past seven days; and the answers to the importance question. The percent of the best scores is calculated as the percent of the patients scoring 50, 75 or 100 on the first questions and 60, 80 or 100 on the second and the third question. A third of the patients stated that their health-related quality of life was worse than one month before the injury. Similarly, 43% of the patients stated that their quality of life after the injury and the surgical treatment was either very poor, poor or fair, while 40% of the patients stated the same about their overall quality of life. The patients regarded the pain, appearance and mood issues as the most important domains during the first postoperative month.

The complicated patients scored significantly lower on all of the three global questions (*p* < 0.01 for all the questions) compared to the uncomplicated group. There were no differences regarding the importance question.

## 4. Discussion

In the study presented herein, unilateral mandibular corpus fractures were most commonly seen in males in the third or fourth decade of life, in accordance with the previous reports [12,13]. The injuries were usually caused by violence or traffic accidents, similar to the reports in the relevant literature [12,14]. Left-sided fractures were more common and were usually caused by a direct blow to the mandible delivered by the right hand of the attacker.

The majority of patients suffered a low degree of posttraumatic pain and a small interincisal distance. The postoperative pain and trismus are related to the inflammatory response associated with the mandibular injury [15]. The degree of symptoms such as edema, pain, and trismus depends on the force delivered to the face and the extent of inflammation associated with the trauma, resulting in varying levels of these clinical manifestations. The facial injury causes an inflammatory response characterized by vasodilation and increased capillary permeability, as well as polymorphonuclear and monocyte infiltration in the injured tissue. These changes take days to be developed. The cellular invasion in the trauma region affects the outcome of fracture treatment, leading to a potential pitfall in the early fracture surgery (the lack of blood supply and inflammatory response may lead to an impaired healing process and an unfavorable outcome) [7].

The incidence of complications (15%) in this study represents a generalizable patient population and treatment method. The present study showed malocclusion occurring in all the complicated patients. Mandible fracture complication rates vary from 7 to 29%, with malocclusion [16] regarded as the most important and one of the most frequent types of complications; as well as infection [17,18]. A study on 273 patients conducted by Furr et al. showed complications in 6.6% of all treated patients (with hardware exposure as the most frequent complication). The only factors that were significantly associated with the complications were smoking, alcohol abuse and the use of plating procedures. However, the influence of the rigid fixation methods is explained in accordance with the fact that they are commonly used in severe and multiple fractures, which are associated with a naturally greater risk of complications [18]. Munante-Cardenas et al. performed a retrospective study on 119 mandibular fracture cases, reporting a complication rate of 30.2%. The most frequent type of complication was an infection (36.1% of all cases with complications), different from the study presented herein (which showed no infectious complications). The authors of the study did not provide information on the presence or absence of antibiotic prophylaxis, which might have contributed to the difference in complication types [14]. Pickrell and Hollier stated that there is no evidence to support the postoperative antibiotic use, while the preoperative antibiotic prophylaxis is beneficial (especially in open fractures occurring in tooth-bearing regions) [16]. The results of the study by Furr et al. showed no association between the antibiotic use and complication [18]. However, their study showed a variable prophylactic regimen. A comparative study on different antibiotic protocols should be conducted in order to assess the need for antibiotic prophylaxis and/or postoperative treatment in the prevention of complications.

A retrospective study performed on 33 patients treated at a level 1 trauma center by Webb et al. showed that there was no significant difference in the incidence of postoperative complications between patients who were treated within 72 h after injury and the patients treated after that period of time [12]. Barker et al. performed a retrospective study on 83 patients with a mean time-to-repair time of 6.7 days, reaching a conclusion that a delay in surgical treatment did not influence the complication rate [19]. However, Biller et al. performed a retrospective study on 84 patients with a conclusion that technical complications (weakness of the marginal mandibular nerve, malocclusion, persistent pain and intraoral dehiscence) were significantly more common in the group of patients with delayed surgery. Different reduction methods (open reduction internal fixation, maxillomandibular fixation or both) did not contribute to a higher risk of complications. There was no significant difference regarding the incidence of infectious complications. The authors concluded that the tissue edema or inflammation might play a role in the complications occurring after delayed treatment, leading to a suggestion that the surgery should be performed during the first 72 h [5]. Luz et al. stated that the treatment delay is associated with complications requiring reoperation. However, the time-to-repair in their study was 13.5 ± 9 days in the uncomplicated and 19.1 ± 18.7 in the reoperated group—markedly longer than in other studies [17]. A systematic review performed by Hermund et al. in 2006 showed that there is no strong evidence to support either acute or delayed treatment, most likely owing to a lack of randomized studies, as well as a significant amount of confounding factors (such as alcohol and substance abuse, non-compliance etc.) which increase the complication risk [7]. A similar conclusion was reached in a systematic review by Hurrell and Batstone conducted in 2014, with an emphasis on the need for randomized controlled trials on this subject [20]. The study performed by Furr et al. also did not show any association of complications with the time-to-repair [19].

A study on 249 patients with a unilateral mandibular fracture showed no significant difference in a complication risk between patients treated by open reduction internal fixation with manual reduction or arch bar placement [13]. In a prospective study on 83 patients treated for unilateral mandibular fractures with maxillomandibular fixation, Anyanechi and Saheeb showed that posttraumatic pain and trismus are associated with the onset of complications [15]. Their study emphasized the influence of inflammation-mediated symptoms on the complications after closed reduction—an approach which is unable to estimate the influence of surgical trauma on postoperative complications. Therefore, a repeated analysis on patients who underwent open reduction with an intraoral approach was conducted in the study presented herein, with a confirmation of the conclusion that the posttraumatic pain and trismus are positively related to the onset of non-infectious complications. The presence of pain in both the complicated and the uncomplicated patients (despite the administration of analgesic drugs) illustrates the rapid inflammatory response to the skeletal injury. Regarding the practical use of this finding, it is necessary to analyze whether delayed surgery after a confirmed reduction of the inflammatory symptoms leads to a lower complication risk in comparison with the surgery performed on symptomatic patients.

The aesthetic and functional disturbances associated with facial trauma often result in social adjustment problems; as well as post-traumatic stress disorder, problems with body image, mood disorders, and overall poor quality of life after the injury and treatment [10]. A multi-centric German study conducted on 600 oral and maxillofacial surgical patients showed that there is a significant degree of treatment anxiety in these patients before the therapeutic intervention [21]. Patients suffering from mandibular fractures usually enter the health care system through emergency care, not allowing them to express mental health (anxiety or depression) issues. In a sample consisting mainly of unmarried men injured during interpersonal violence, Gironda et al. found that post-treatment depression was significantly associated with postsurgical pain and oral health problems during healing [9]. The present study showed that a third of the patients suffered a reduction in the health-related quality of life after surgery compared to one month before the injury. Furthermore, nearly half of the patients identified their health-related and overall quality of life as ‘fair’ or worse in a week after treatment. Chewing was affected the most, and the majority of patients stated that pain was the most important domain of the questionnaire. These findings suggest that it is necessary to address the postoperative pain and chewing issues during rehabilitation, as well as to organize psychological support to the operated patients.

This study has several limitations. A relatively small sample was treated with a single treatment method on the same postoperative day. The small size of the sample might influence the statistical validity of certain associations. Furthermore, an analysis of patient compliance was not performed. Regarding the issue of noncompliant patients, the surgical decision to advocate more aggressive treatment approaches (such as internal rigid fixation) is often noted as an attempt to decrease the risk of repeated injury. Zazzali et al. stated that healthcare professionals dealing with mandibular fractures (oral, maxillofacial and otolaryngology surgeons) identify homelessness, alcohol and drug abuse as the most important factors contributing to patient noncompliance [22]. Given the fact that the facial skeleton surgery is most commonly performed due to trauma from motor vehicle accidents or violence [23]—and that systemic disease may also affect the mandible and the temporomandibular joint, requiring treatment [24]—future studies should investigate any potential differences in the psychological outcomes of injured and non-injured patients undergoing mandibular surgery. Further analysis should span on multiple treatment methods with varying time-to-repair in order to analyze the effect of those variables on the relationship between the inflammatory symptoms and the incidence of complications.

## 5. Conclusions

Facial skeleton injuries are widely researched due to the increases in incidence and the amount of traumatic force, as well as the aesthetic requests delivered by the patients. The study presented herein showed that the amount of preoperative pain and trismus may be positively related to the onset of complications occurring after the rigid fixation of mandibular corpus fractures. Furthermore, the postoperative health-related and overall quality of life was unsatisfactory in nearly half of the patients, presenting a need for psychological intervention during rehabilitation. The patients suffering from complications showed a significantly lower quality of life regarding appearance, swallowing and anxiety. Given the fact that the modern lifestyle is associated with an increased risk of facial trauma, it is imperative to acknowledge the potential risk factors for postoperative complications. Additionally, this study should inspire the clinicians to provide psychological support to the patients in order to achieve a superior therapeutic effect.

## Figures and Tables

**Table 1 medicina-55-00109-t001:** Distribution of patients by age and gender.

Patients’ Age, Years	Male	Female	Total
21–30	17 (28.3)	3 (5.0)	20 (33.3)
31–40	15 (25)	3 (5.0)	18 (30.0)
41–50	10 (16.7)	1 (1.7)	11 (18.3)
51–60	5 (8.3)	2 (3.3)	7 (11.7)
61–70	4 (6.7)	0 (0.0)	4 (6.7)
Total	51 (85.0)	9 (15.0)	60 (100.0)

Values are number (percentage).

**Table 2 medicina-55-00109-t002:** Distribution of complications in relation to the intensity of preoperative facial pain and trismus.

**Pain Intensity**	**Patients with Complications**	**Patients without Complications**	**Total**
0–2.5	0 (0.0)	25 (41.7)	25 (41.7)
2.6–5.0	3 (5.0)	15 (25.0)	18 (30.0)
5.1–7.5	2 (3.3)	7 (11.6)	9 (15.0)
7.6–10.0	4 (6.7)	4 (6.7)	8 (13.3)
Total	9 (15.0)	51 (85.0)	60 (100.0)
**Mouth Opening (mm)**	**Patients with Complications**	**Patients without Complications**	**Total**
<5	4 (6.6)	13 (21.7)	17 (28.4)
6–10	3 (5.0)	9 (15.0)	12 (20.0)
11–15	1 (1.7)	13 (21.7)	14 (23.3)
16–20	1 (1.7)	11 (18.3)	12 (20.0)
20–25	0 (0.0)	5 (8.3)	5 (8.3)
Total	9 (15.0)	51 (85.0)	60 (100.0)

Values are number (percentage).

**Table 3 medicina-55-00109-t003:** The frequency of different types of postoperative complications and therapeutic strategies.

**Complication Type**	**Value**
Occlusal derangement	9 (100)
Facial asymmetry	7 (77.8)
Impaired mouth opening (<35 mm)	4 (44.4)
Malunion	2 (22.2)
Nonunion	2 (22.2)
**Treatment Type**	**Value**
Counseling	9 (100)
Physiotherapy	7 (77.8)
Nonsteroidal anti-inflammatory drugs	4 (44.4)
Intermaxillary fixation	2 (22.2)
Refracture	2 (22.2)

Values are numbers (percentage from the total number of complicated patients).

**Table 4 medicina-55-00109-t004:** The University of Washington Quality of Life questionnaire domain scores and the number of respondents to each domain.

	Quality of Life							Mean Score	% of the Best Score
Domains	0	25	33	50	67	75	100		
Pain	6	14		8		12	20	60.83	33.3
Appearance	2	8		22		18	10	60.83	16.6
Activity	8	4		18		12	18	61.67	30.0
Recreation	10	8		12		16	14	56.67	23.3
Swallowing	8		12		20		20	62.27	33.3
Chewing	16			28			16	50.0	26.6
Speech	6		8		16		30	72.27	50.0
Shoulders	12		10		18		20	58.93	33.3
Taste	8		6		14		32	72.27	53.3
Saliva	10		8		14		28	66.70	46.67
Mood	4	6		18		18	14	63.33	30.0
Anxiety	12		10		18		20	58.93	33.3

**Table 5 medicina-55-00109-t005:** The global questions, the importance question in the University of Washington Quality of Life questionnaire and the number of respondents to each question/domain.

**The Global Questions**											
**Domains**	**0**	**20**	**25**	**40**	**50**	**60**	**75**	**80**	**100**	**Mean Score**	**% of the Best Scores**
Health-related QOL (compared to 1 month before injury)	10		10		12		10		18	56.67	66.67
Health-related QOL (past week)	8	6		12		12		16	6	53.33	56.67
Overall QOL (past week)	8	4		12		14		16	6	54.67	60.0
**The importance question**											
**Domains**	**Number of patients choosing the domain**									**Rank**	
Pain	26									1=	
Appearance	26									1=	
Mood	20									3	
Activity	16									4	
Recreation	12									5=	
Swallowing	12									5=	
Chewing	6									7=	
Anxiety	6									7=	
Shoulders	2									9	
Speech	0									10=	
Taste	0									10=	
Saliva	0									10=	

QOL—quality of life.

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
