# Peer review of "The Quality of Life of Patients with Surgically Treated Mandibular Fractures and the Relationship of the Posttraumatic Pain and Trismus with the Postoperative Complications: A Prospective Study"

_medicina, 2019, doi:10.3390/medicina55040109_

Round 1

Reviewer 1 Report

Dear Authors, scientific study is certainly of great interest. Having associated with the medical evaluation of the patient also an evaluation of his quality of life, gives originality to the work. There are some changes to be made for the work to be complete for the scientific journal.

-In the "intruduction" section it is necessary to correct the phrase "Being the only movable bone" with a sentence in medical language (lane 37).

-Furthermore in the "introduction" section for completeness it is necessary to add references to the following scientific study, and some information about pharmacological protocols after mandibular surgery:

Troiano, G., Laino, L., Cicciù, M., Cervino, G., Fiorillo, L., D’Amico, C., Zhurakivska, K., Muzio, L.L. Comparison of two routes of administration of dexamethasone to reduce the postoperative sequelae after third molar surgery: A systematic review and meta-analysis (2018) Open Dentistry Journal, 12, pp. 181-188

-In the "Materials and methods" section it is necessary to create a substructure to the text with subsections:

for example, create a section with a list for "inclusion and exclusion criteria"; create a sample section; create a section where the individual analysis methods used (oral hygiene, degree of pain and trismus, mouth opening index, VW-QoL v4) are analyzed. Please specify for each of these a small description in the subsection.

-In the "materials and methods" section specify the type of analgesic used.

-Results section is okay. 

-Please enlarge the "discussion" section by adding the following references for completeness of the work about facial fractures and TMJ disorders:

Runci M, De Ponte FS, Falzea R, Bramanti E, Lauritano F, Cervino G, Famà F, Calvo A, Crimi S, Rapisarda S, Cicciù M. Facial and Orbital Fractures: A Fifteen Years Retrospective Evaluation of North East Sicily Treated Patients. Open Dent J. 2017 Oct 31;11:546-556. doi: 10.2174/1874210601711010546. eCollection 2017.

Isola, G., Ramaglia, L., Cordasco, G., Lucchese, A., Fiorillo, L., Matarese, G. The effect of a functional appliance in the management of temporomandibular joint disorders in patients with juvenile idiopathic arthritis (2017) Minerva Stomatologica, 66 (1), pp. 1-8.

-Enlarge 2-3 lines conclusion specifying the medical epidemiological value of the study.

Confident in your work, 

Best Regards.

Author Response

Dear Anonymous Reviewer #1, 

First of all, thank you for your kind and useful comments.

We are sending attached the letter of response (point-by-point) to your remarks.

Kind regards, 
Corresponding author

Reviewer 2 Report

The manuscript is well thought out and organized. The authors did a good job presenting the study and background associated. 

However, there are important English mistakes along the manuscript that may, in some instances, difficult the reading and understanding of the information. A minor grammar revision should be done before publication. The authors can also consider adding more recent references, especially between the years 2016 to 2019 (they are lacking). Overall the manuscript has merit and should be considered for publication.

Author Response

Dear Anonymous Reviewer #2, 

First of all, thank you for your kind and useful comments.

We are sending attached the letter of response (point-by-point) to your remarks.

Kind regards, 
Corresponding author

Reviewer 3 Report

Dear Authors,

Thanks for submitting manuscript, which presents an interesting prospective study on the treated mandibular fractures with pain and complications.

First, the paper is quite interesting, my comments for improvement are focused mainly on the introduction. 

I do not clearly understand purpose of study (correlation between complications & the quality of life). I think better explaining of study's aim is required for clarity of study design and analysis of factors (pain intensity, mouth opening, bite force, muscle force, etc). 

Second, I think the number of patients with complication is not enough for pain intensity or mouth opening related to the onset of complications. How to explain the amount of preoperative pain of the patients without complication?
additionally, how to calculate the sample size? if you estimated the required sample size, you need to explain in manuscript including the approval of the institutional review board.

third, I recommend that evaluation of QOL(quality of life) divided with/without complications.

Author Response

Dear Anonymous Reviewer #3, 

First of all, thank you for your kind and useful comments.

We are sending attached the letter of response (point-by-point) to your remarks.

Kind regards, 
Corresponding author

Round 2

Reviewer 1 Report

Dear Authors,

thank You for accepting my suggestions, now i think that the work is ready for publication. 

Author Response

Dear Reviewer, 

thank you for your kind comment.

Sincerely yours, 

Reviewer 3 Report

Dear Authors, 

Thanks for the revision efforts.

Congratulations for performing the correction so soon in an excellent way to the suggestions. 

Author Response

(The authors gave the same response as above.)
